# Diagnostic Performances of Urinary Methylmalonic Acid/Creatinine Ratio in Vitamin B12 Deficiency

**DOI:** 10.3390/jcm9082335

**Published:** 2020-07-22

**Authors:** Sopak Supakul, Floris Chabrun, Steve Genebrier, Maximilien N’Guyen, Guillaume Valarche, Arthur Derieppe, Adeline Villoteau, Valentin Lacombe, Geoffrey Urbanski

**Affiliations:** 1Department of Internal Medicine, University Hospital, 49000 Angers, France; 140521ms@tmd.ac.jp (S.S.); nguyen.maximilien@gmail.com (M.N.); guillaume.valarche@sfr.fr (G.V.); arthurderieppe@gmail.com (A.D.); adeline.villoteau@hotmail.fr (A.V.); lacombe.valentin31@gmail.com (V.L.); 2Department of Biochemistry, University Hospital, 49000 Angers, France; Floris.chabrun@chu-angers.fr (F.C.); steve.genebrier@hotmail.fr (S.G.); 3Department of Immunology, University Hospital, 35000 Rennes, France

**Keywords:** vitamin B12 deficiency, plasma homocysteine, plasma methylmalonic acid, urinary methylmalonic acid, biomarker, renal impairment

## Abstract

Sole measurement of plasma vitamin B12 is no longer enough to identify vitamin B12 (B12) deficiency. When plasma vitamin B12 is in the low-normal range, especially between 201 and 350 ng/L, B12 deficiency should be assessed by measurements of plasma homocysteine and/or plasma methylmalonic acid (MMA). However, these biomarkers also accumulate during renal impairment, leading to a decreased specificity for B12 deficiency. In such cases, urinary methylmalonic acid/creatinine ratio (uMMA/C) could be of interest, due to the stable urinary excretion of MMA. The objectives were to evaluate the influence of renal impairment on uMMA/C compared to plasma homocysteine and plasma methylmalonic acid, and to determine the diagnostic performances of uMMA/C in the diagnosis of B12 deficiency. We prospectively studied 127 patients with a plasma B12 between 201 and 350 ng/L. We noticed that uMMA/C was not dependent on renal function (*p* = 0.34), contrary to plasma homocysteine and plasma methylmalonic acid. uMMA/C showed a perspective diagnostic performance (AUC 0.71 [95% CI: 0.62–0.80]) and the threshold of 1.45 umol/mmol presented a high degree of specificity (87.9% [95% CI: 72.0–98.9]). In conclusion, uMMA/C is a promising biomarker to assess vitamin B12 status in doubtful cases, notably during renal impairment.

## 1. Introduction

Vitamin B12 (B12, or cobalamin) deficiency is a frequent condition that affects 1.5% of the general population [1,2], and up to 10–15% of subjects above 60 years old [3]. The early diagnosis and treatment of B12 deficiency is important to cure the incipient disease and prevent the development of irreversible neurological disorders [4]. For decades, the status of B12 deficiency was only based on the plasma B12 measurement, with a threshold around 200 ng/L (148 pmol/L) [4]. Nevertheless, plasma B12 does not correctly reflect the intracellular cobalamin status. Indeed, typical clinical and biological signs of B12 deficiency can be observed in patients with plasma B12 over 200 ng/L, notably between 201 and 350 ng/L [4,5,6,7]. As a consequence, the interval of plasma B12 between 201 and 350 ng/L defines a “grey zone” where the B12 status remains uncertain.

To identify such latent B12 deficiency, different biomarkers have been proposed based on physiological B12 functions. B12 is involved as a cofactor for two enzymes, methionine synthase [8], and methylmalonyl-CoA mutase [9], with homocysteine (HCyst) and methylmalonic acid (MMA) as two respective substrates. In the case of B12 deficiency, HCyst and MMA become accumulated [10]. Thus, these two metabolites allow a functional assessment of the B12 status, and the increases of total plasma HCyst and plasma MMA are frequently used to confirm a B12 deficiency in the case of serum/plasma B12 between 201 and 350 ng/L [10,11].

Although plasma HCyst and MMA show high sensitivity for clinical deficiency (96% and 98%, respectively), some authors pointed out problems of specificity as both markers are affected by factors irrelevant to B12 deficiency [4]. Indeed, different conditions may increase the plasma HCyst levels, like folate deficiency [12,13], advanced age [14], and renal failure [15,16]. Plasma MMA is more specific to the B12 status than plasma HCyst, but an increase of plasma MMA level is also observed at a decreased kidney function [17,18]. Thus, the diagnosis of B12 deficiency is difficult to establish in the case of renal impairment.

MMA is excreted by kidneys, and its urinary concentration makes it a sensitive marker of cellular B12 depletion [19]. The ratio of uMMA on urinary creatinine value (uMMA/C) could limit the influence of the renal impairment [20]. Some previous studies argued for the urinary MMA (uMMA) as a biomarker of B12 deficiency, as it is correlated with plasma MMA and symptoms of B12 deficiency [21,22,23,24]. However, none of these studies could explicitly demonstrate the advantageous diagnostic performances of urinary MMA for detection of B12 deficiency. Our objectives were (i) to evaluate the influence of kidney function on uMMA/C, compared to plasma HCyst and plasma MMA; (ii) to determine the diagnostic performances of uMMA/C in patients with different renal status and suspected of being B12-deficient.

## 2. Materials and Methods

### 2.1. Ethics

This study was approved by the Bioethical Committee of Angers University Hospital (*n*° 2015-15). We applied the Strengthening the Reporting of Observational studies in Epidemiology (STROBE) statement to observational studies. All patients gave informed consent. The study was conducted in accordance with the Declaration of Helsinki.

### 2.2. Study Population

We prospectively investigated B12 status of all inpatients hospitalized in the department of internal medicine, Angers University Hospital, with the exception of patients who did not require blood samples testing. Recruitment was carried out for a period of 15 weeks. The present study included all patients with plasma B12 between 201 and 350 ng/L. Patients with missing data for plasma B12, plasma HCyst, plasma MMA and/or uMMA/C were excluded.

### 2.3. Data Collection

The following general data and biochemical profiles were collected: age, gender, body mass index (BMI), alcohol and tobacco misuse, estimated glomerular filtration rate (eGFR) evaluated with the MDRD (Modification of the Diet in Renal Disease) formula [25], plasma B12, plasma folate, plasma HCyst, plasma MMA and uMMA, and urinary creatinine. 

### 2.4. Biological Assays

#### 2.4.1. Plasma Vitamin B12 

The plasma vitamin B12 was determined by competitive chemiluminescent immunoassay on ADVIA Centaur^®^ system (Siemens Healthcare Diagnostics Inc., Tarrytown, NY 10591-5097, USA). The coefficient of variation was 1.3–4.1%.

#### 2.4.2. Plasma Homocysteine 

The plasma homocysteine was determined by HPLC-MS/MS (Agilent 1200 Infinity Series, Agilent Technologies, Santa Clara, CA, USA; Triple Quad™ 4500, SCIEX, Framingham, MA, USA). The coefficient of variation was of 1.5–1.6%.

#### 2.4.3. Plasma Methylmalonic Acid 

The plasma methylmalonic acid was determined by HPLC-MS/MS (Agilent 1200 Infinity Series, Agilent Technologies, Santa Clara, CA, USA; Triple Quad™ 5500, SCIEX, Framingham, MA, USA). The coefficient of variation was of 2.6–5.0%.

#### 2.4.4. Urinary Methylmalonic Acid

Urine samples were collected in sterile tubes and stored at −20 °C until extraction and analysis. Urinary methylmalonic acid was determined by GC-MS (QP 2010S, Shimadzu, Kyoto, Japan). The coefficient of variation was 7.25–8.8%.

Details about measurements of plasma B12, plasma HCyst, plasma MMA, and urinary uMMA are available in the Appendix A.

### 2.5. Definition for Vitamin B12 Deficiency

Patients with B12 deficiency with plasma B12 ≤200 ng/L were not included in this study because we were only interested in patients with plasma B12 in the “grey zone” (201–350 ng/L). As a consequence, B12 deficiency was defined as a plasma B12 between 201–350 ng/L combined with elevated levels of (i) plasma HCyst (≥13 µmol/L for women [10,26], ≥15 µmol/L for men [10,26], ≥20 µmol/L for patients older than 65 years old [14], but not considered in case of plasma folates <4 µg/L [27]) or (ii) plasma MMA (≥0.35 µmol/L [28,29]), as recommended by experts [4,11]. 

### 2.6. Statistical Analysis

Qualitative data were expressed as absolute values and percentages. Quantitative data were expressed as medians and quartiles. 

Regression curves were used to test the relation between eGFR and plasma MMA, plasma HCyst, and uMMA/C by the F test. To improve the representation of this relationship, linear regression, semilog curves and hyperbolic curves were compared and selected according to the method of Akaike’s Information Criterion (AIC) [30]. The ratio of uMMA/C represents two fluctuating variables, therefore the dependence of y = uMMA/C on x = eGFR was presented after logarithmic transformation (log10(y + 1)) of y-axis to suppress deviation of “y” from normality. Diagnostic performances were evaluated by receiver operating characteristic (ROC) curves with the calculation of the area under the curve (AUC). The best threshold was selected using the Youden’s index. The alpha risk was set at 5%. The AUC were presented with their 95% confidence interval (95% CI). The analyses were performed using SPSS v24.0 (IBM Corp, New York, NY, USA) and GraphPad Prism v6.01 softwares (GraphPad Software, San Diego, CA, USA). 

## 3. Results

### 3.1. The Study Population

Among the 490 patients tested for B12 status, 144 (29.4%) had plasma B12 in the range 201–350 ng/L, of whom 127 patients had no missing data. The flowchart is detailed in Figure 1.

The characteristics for study population are summarized in Table 1. The whole population had a median age of 66 years (44–81), with 77/127 (60.6%) women. Among the 127 patients included in the study, 69 (54.3%) had a B12 deficiency according to plasma MMA and/or HCyst measurements. 

### 3.2. Modeling of Relation between the Creatinine Clearance and the Biochemical Parameters of Interest

Semilog linear regression curves better approximated the chart of plasma HCyst vs. eGFR than linear regression lines did. Hyperbolic better approximated the chart of plasma MMA vs. eGFR in comparison to linear fit. The ratio of probabilities according to the AIC method were as follows: 2.01 for plasma HCyst for women, 4.28 for plasma HCyst for men, and 1.20 for plasma MMA compared to linear regression. Difference of fit for uMMA/C was in favor of semilog linear regression, performed after a logarithmic transformation (log10(y + 1)) (ratio of probabilities according to AIC method: 1.97).

### 3.3. Influence of the Renal Function on Plasma Homocysteine, Plasma Methylmalonic Acid and Urinary Methylmalonic Acid to Creatinine Ratio

Plasma HCyst and plasma MMA were dependent on eGFR (*p* = 0.02 and *p* = 0.004, respectively) while uMMA/C was independent of eGFR (*p* = 0.34) (Figure 2). On the non-linear regression curves, the corresponding value for thresholds for plasma HCyst (13 and 15 µmol/L) and for plasma MMA (0.35 µmol/L) on the x-axis appeared near 100 mL/min/1.73 m^2^ for eGFR: 102 for plasma HCyst for women, 108 for plasma HCyst for men and 93 for plasma MMA (Figure 2).

### 3.4. Diagnosis Performances for Plasma B12, Plasma Homocysteine, Plasma Methylmalonic Acid and Urinary Methylmalonic Acid to Creatinine Ratio

Among the whole population, AUC for B12 deficiency diagnosis was 0.52 [95% CI: 0.42–0.63] for plasma B12 (*p* = 0.65) (Figure 3A), 0.85 [95% CI: 0.78–0.91] for plasma HCyst (*p* < 0.0001) and 0.90 [95% CI: 0.85–0.96] for plasma MMA (*p* < 0.0001). However, it is worth noting that ROC analysis of the diagnostic performance for plasma MMA and HCyst cannot be considered as accurate because the same markers were used to stipulate B12-deficiency. AUC for uMMA/C ratio was 0.71 [95% CI: 0.62–0.80] (*p* < 0.0001) (Figure 3B). 

The threshold of uMMA/C ratio with the higher Youden’s index (0.43) was 1.45 µmol/mmol. For this threshold, the sensitivity was 55.1% [95% CI: 42.6–67.1] and the specificity was 87.9% [95% CI: 76.7–95.0].

As on the semilog curves (Figure 2), the corresponding value for thresholds for plasma HCyst (13 and 15 µmol/L) and for plasma MMA (0.35 µmol/L) on the x-axis appeared near 100 mL/min/1.73 m^2^ for eGFR, so we focused on patients with eGFR <100 mL/min/1.73 m^2^ (*n* = 64/127). In those patients, AUC for B12 deficiency diagnosis of uMMA/C ratio was 0.72 [95% CI: 0.60–0.85] (*p* = 0.003). With the threshold of 1.45 µmol/mmol, the sensitivity was 48.8% [95% CI: 32.9–64.9] and the specificity was 91.3% [95% CI: 72.0–98.9].

## 4. Discussion

Plasma B12 measurement is not sufficient to precisely evaluate the B12 status. Patients with subnormal plasma B12 values may have clinical and biological features associated with B12-deficiency [5,6,7,31,32]. Experts recommend measuring other biomarkers such as plasma HCyst and plasma MMA to assess the diagnosis of B12 deficiency in the case of low-normal values of plasma B12 (200–350 ng/L) [4,11,33]. Nevertheless, these biomarkers are not strictly specific for B12 deficiency: vitamin B2, B6, and folate deficiencies can raise the plasma HCyst levels [34,35,36], while dehydration and increased levels of propionic acid can also increase the plasma MMA levels [37]. However, the main impediment is renal impairment as it increases both plasma HCyst and plasma MMA values, leading to an important loss of specificity in this situation [15,16,17,18,38,39,40,41,42]. Fedosov et al. proposed an interesting index calculation based on measurements of different biomarkers to suppress the “inherent noise” of each individual marker of B12-deficiency [43,44]. However, this index was not studied in patients with renal failure, since patients with blood creatinine >100 µmol/L were excluded from these studies. As a consequence, the diagnosis of B12 deficiency may be difficult in the case of renal impairment. The search for markers independent of renal function is essential to avoid the overdiagnosis of B12 deficiency.

Previous reports suggested that urinary MMA could become a good diagnostic marker of B12 deficiency due to its good stability in urine [19], its association with cellular B12 deficiency [45], and the absence of expected decrease of its urinary excretion before severe renal failure [24]. Previous studies attempted to demonstrate, albeit indirectly, the diagnostic value of uMMA by showing its correlation with plasma MMA, for example [21,22]. Yet, most of them excluded patients with renal failure [21,22]. For the first time, our study aimed at comparing the influence of renal impairment on plasma HCyst and plasma MMA levels, and on the uMMA/C ratio, as well as at testing the diagnostic performances of the uMMA/C ratio to assess the B12 status.

Firstly, this study confirmed both the considerable occurrence of B12 deficiency in a hospitalized population (89/490, 18.2%) and the high proportion of patients with B12 deficiency among patients with B12 between 200 and 350 ng/L (69/89, 77.5%).

Moreover, plasma B12 failed to diagnose B12 deficiency in the study’s range from 201 to 350 ng/L. These data highlighted the importance of the measurements of functional biomarkers in doubtful cases in order to prevent the development of hematological and neurological disorders.

We observed a link between both plasma HCyst and plasma MMA and renal function. Interestingly, these biomarkers increased at early stages of renal impairment with eGFR <100 mL/min/1.73 m^2^. Van Loon et al. already reported similar results for plasma MMA at the early stage of renal impairment (≤90 mL/min/1.73 m^2^) [18]. The impairment of renal function could affect the interpretation of the plasma HCyst and the plasma MMA. This awakes some concerns because a slight renal impairment is frequent in inpatients, notably in older populations that are affected by B12 deficiency [42]: in our study, 50.4% of patients presented eGFR <100 mL/min/1.73 m^2^. Aparicio-Ugarriza et al. already suggested to set different cut-offs according to age and gender [46], as we did for plasma HCyst in the present study. This solution has the advantage of using the same biomarker. However, the link between eGFR and those biomarkers does not seem to be linear but rather a logarithm derivate. This non-linear relation makes it difficult to establish an easily usable calculation in daily practice, adapting the level of biomarker to the level of renal impairment. The other solution could be to identify a second-line biomarker, which is independent of renal status and holds a high degree of specificity.

Previous reports suggested that urinary MMA could be a perspective diagnostic marker of B12 deficiency [19,45], notably due to the absence of decreased excretion before severe renal failure [24]. In our study, we demonstrated that uMMA/C was independent of renal function. Moreover, with a threshold of 1.45 µmol/mmol, the uMMA/C ratio was associated with good diagnostic performances for B12 deficiency as a second line assay (sensitivity 55.1%, specificity 87.9%). This ratio can be considered as a biomarker of interest for detection of B12 deficiency, especially in patients with eGFR <100 mL/min/1.73 m^2^. The threshold identified in our study (1.45 µmol/mmol) is close to the threshold suggested by Flatley et al. (1.50 µmol/mmol). The value is applicable in a study population with plasma B12 between 201 and 350 ng/L, i.e., the range where the use of other biomarkers is recommended for the detection of B12 deficiency [4,10]. In contrast, it is not recommended to perform any additional measurements by other biomarkers in patients with plasma B12 ≤200 ng/L and those with plasma B12 >350 ng/L. Thus, we found it unnecessary to assess the diagnostic performances in patients for whom the diagnosis was already confirmed or excluded based on the assay of plasma B12.

In the absence of a gold standard for B12 deficiency, we used the algorithm recommended by experts to diagnose B12 deficiency, according to plasma HCyst and plasma MMA [11]. Since folate deficiency itself may increase the plasma HCyst level, plasma HCyst was not considered in the case of folate <4 µg/L to avoid overdiagnosis of B12 deficiency [27]. The higher values of plasma HCyst are observed more frequently in men than in women [47] and even more in elderly patients [14,48]. This explains the different recommended cut-offs of plasma HCyst to define B12 deficiency [10,14,26]. The best threshold of plasma MMA is less consensual, varying from 0.28 to 0.40 among studies [28,29,49,50,51].

As the elevations of plasma HCyst and plasma MMA were included in our definition of B12 deficiency, this stipulation artificially increased their diagnostic performances, contrary to those of uMMA/C. This explains why we did not compare the diagnostic performances of these biomarkers and they were only present as the descriptive results. Moreover, as the plasma HCyst and plasma MMA are dependent on renal function, we cannot exclude that the prevalence of B12 deficiency in our study was overestimated in patients with renal impairment. Indeed, we noted lower values of eGFR in the subgroup of patients with B12 deficiency. Unfortunately, in the absence of an external gold standard, there was no perfect solution to classify the patients.

In conclusion, this study demonstrated that urinary MMA to urinary creatinine ratio can be a perspective biomarker to assess vitamin B12 deficiency. The uMMA/C ratio was independent of the renal function, contrary to the other reference biomarkers of B12 deficiency, plasma HCyst and plasma MMA, that could be misinterpreted in the case of renal impairment, even at its early stage. uMMA/C presented good diagnostic performances for B12 deficiency, and notably a high level of specificity (87.9% [95% CI: 76.7–95.0]) at a threshold of 1.45 µmol/mmol in patients with subnormal values of plasma B12 (201–350 ng/L). The uMMA/C ratio appeared to be a useful tool to confirm a diagnosis of B12 deficiency in doubtful cases, particularly for renal impairment.

## Figures and Tables

**Figure 1 jcm-09-02335-f001:**
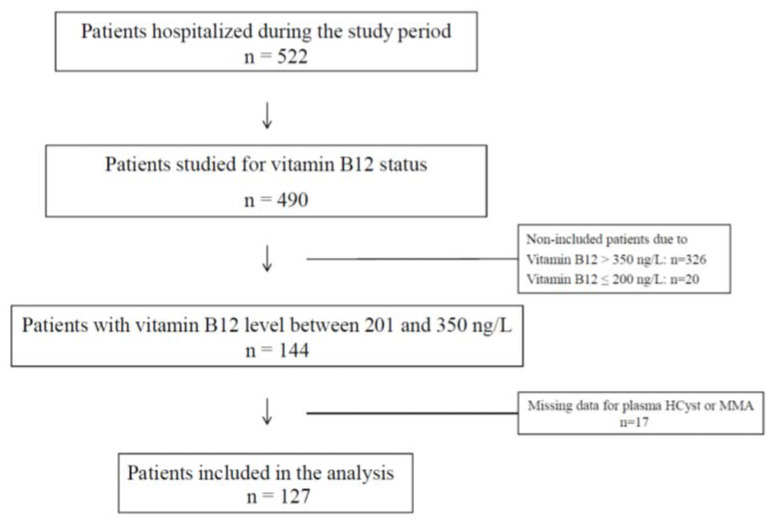
Flow chart of the study population.

**Figure 2 jcm-09-02335-f002:**
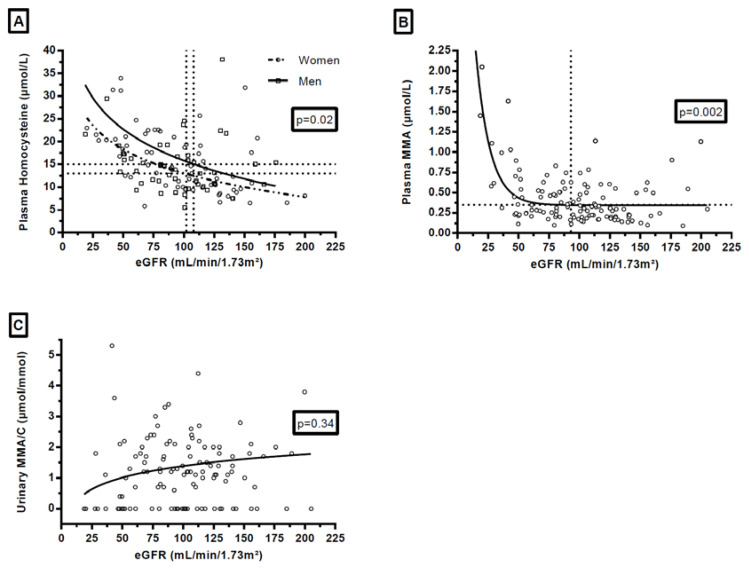
Influence of renal function on plasma homocysteine (**A**) in women and in men, plasma methylmalonic acid (**B**) and urinary methylmalonic acid to creatinine ratio (**C**). Footnotes: *p*-values on the graphs referred to the results of the F test on slopes. For enhanced clarity, y-axis was releveled, and 3 plasma HCyst values (56.3, 93.6, and 129 µmol/L) and 1 uMMA/C value (27.1 µmol/mmol) were outside the graphs. The values which correspond to the horizontal dotted lines of the graphs are: 13 µmol/L for graph A for women, 15 µmol/L for graph A for men, and 0.35 mmol/L for the graph B. The values which correspond to the vertical dotted lines of the graphs are: 102 mL/min/1.73 m² for graph A, 108 mL/min/1.73 m² for graph A, and 93 mL/min/1.73 m² for the graph B.

**Figure 3 jcm-09-02335-f003:**
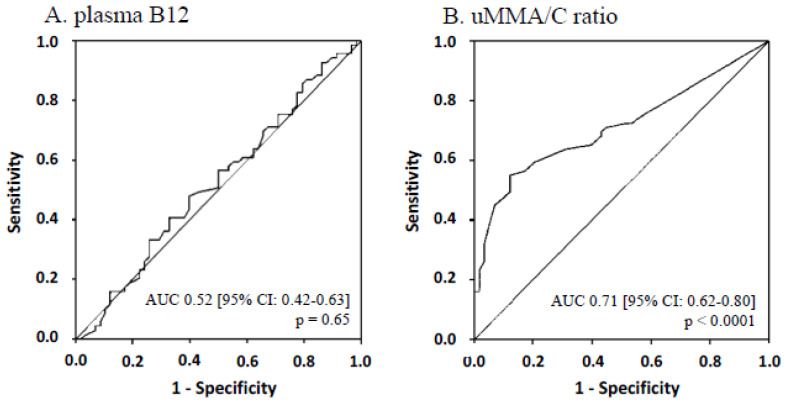
ROC curves and AUC of plasma B12 (**A**) and urinary methylmalonic acid to creatinine ratio (**B**) for the whole population.

**Table 1 jcm-09-02335-t001:** Demographic and biological characteristics of the study population.

Characteristics	Whole Population	B12 Deficiency	No B12 Deficiency
	(*n* = 127)	(*n* = 69)	(*n* = 58)
General characteristics			
Age (years)	66 (44–81)	74 (55–83)	60 (39–73)
Gender (female)	77 (60.6%)	48 (69.6%)	29 (50%)
BMI (kg/m²)	25.1 (22.3–28.5)	25.5 (22.4–29.0)	24.7 (22.4–28.0)
Tobacco misuse	36 (28.3%)	17 (24.6%)	19 (32.8%)
Alcohol misuse	8 (6.3%)	6 (8.7%)	2 (3.4%)
Biochemical Profiles			
eGFR MDRD (mL/min/1.73 m^2^)	99.5 (70.1–123.7)	88.1 (52.9–113.2)	105.3 (81.2–127.0)
Plasma B12 (ng/L)	291 (260–314)	291 (261–315)	290 (260–310)
Plasma Folate (µg/L)	5.6 (3.7–7.4)	4.7 (3.5–7.0)	5.85 (4.20–7.93)
Plasma MMA (mmol/L)	0.32 (0.22–0.54)	0.53 (0.38–0.71)	0.23 (0.18–0.28)
Plasma HCyst (µmol/L)	14.7 (10.7–19.3)	19.1 (14.4–22.5)	11.2 (9.4–14.2)
Urinary MMA (µmol/L)	8.3 (0.0–14.9)	10.1 (0.0–15.2)	5.9 (0.0–13.9)
Urinary Creatinine (mmol/L)	6.5 (3.9–10.2)	6.1 (3.9–8.3)	7.8 (4.2–12.1)
uMMA/C ratio (µmol/mmol)	1.1 (0.0–1.8)	1.6 (0.0–2.2)	0.7 (0.0–1.2)

Qualitative data expressed as absolute values and percentages. Quantitative data expressed as medians and quartiles. BMI = Body Mass Index; eGFR = estimated Glomerular Filtration Rate; MDRD = Modification of Diet in Renal Disease; MMA = Methylmalonic Acid; HCyst = Homocysteine; uMMA/C = urinary Methylmalonic Acid to Creatinine ratio.

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
