# Peer review of "Diagnostic Performances of Urinary Methylmalonic Acid/Creatinine Ratio in Vitamin B12 Deficiency"

_jcm, 2020, doi:10.3390/jcm9082335_

Round 1
Reviewer 1 Report
General comments
The submitted manuscript by Supakul et al “Diagnostic performances of Urinary Methylmalonic Acid/Creatininuria ratio in vitamin B12 deficiency” addresses application of this metabolic marker to correctly assess the B12-status. Blood level of methylmalonic acid (MMA) is affected by both B12-status and kidney function, and the authors suggest to use the ratio of urinary MMA and creatinine (Cr) instead to compensate for renal failure and more clearly expose the effect of B12. The work presents relevant materials, which have a potential to become useful. Yet, the manuscript also suffers from several shortcomings. First of all, there is a number of publications, where this subject has already been examined and discussed in details (see Point of major concern 1). Therefore, the novelty of the submitted manuscript is rather limited. Simultaneous determination of plasma markers B12, MMA and Hcy (apart from urinary MMA/Cr ratio) gives some advantage to the submitted work, because the previous publications measured these plasma markers individually. Yet, Supakul et al could not extract all potential from this combination (Point of major concern 2) and on several occasions present their data incorrectly (Points of major concern 3 and 4). Finally, the authors neither cite nor discuss the previously published works with a direct relevance to the current results (Point of major concern 1). Based on the aforementioned evaluation, I can suggest either major revision or rejection of the manuscript with a possibility to resubmit it (after the raised issues are properly addressed).
Points of major concern
(1) Lines 53-56. Application of urinary MMA (related to urinary creatinine) as a marker of B12-deficiency is well documented. A very similar approach was used by several other groups, and I would suggest to cite and discuss the below references: doi:10.1016/S0009-9120(03)00033-X; doi: 10.1016/j.nut.2004.06.001; doi: 10.1155/2014/921616; PMID: 2591043. The major advantage of the submitted work is a simultaneous measurement of three plasma markers (B12, MMA and Hcy), where two latter metabolites were used to stipulate the B12-status. This fact should be stressed and properly used in the manuscript, see Point of major concern 2.
(2) Lines 100-104. Definition of B12-deficiency is very unclear. The authors should clearly state something like "B12-deficiency was specified as the concurrently high levels of Hcy >= 15 μmol/L and MMA >= 0.35 μmol/L for men ... etc. ", if exactly this approach was used for stratification into “Normal” and “Deficient” groups (which does not follow from the current text). Moreover, I would suggest to combine serum/plasma B12, MMA and Hcy within one diagnostic index (10.1515/cclm-2014-0818) and use it to stratify the data and assess the diagnostic performance of urinary uMMA/Cr ratio. The combined indicator of B12-status (cB12) is becoming a popular tool to suppress the “inherent noise” of each individual marker of B12-deficiency (reviewed in e.g. doi: 10.1093/jn/nxy201; doi: 10.3389/fmolb.2016.00027). Such test of diagnostic performance of uMMA/Cr has never been used before, while the individual blood markers were applied to uMMA on numerous occasions (see comment 1).
(3) Lines 141-155 (Figure 2). Presentation of the data in Fig. 2B and 2C is not accurate. For example, the points in the upper left corner of Fig. 2B considerably deviate upward from the fitting curve. I would suggest using another fitting function (e.g. hyperbolic), to expose a rather steep increase in plasma MMA at eGFR < 50 mL/min/1.73. Fig. 2C shows uMMA/Cr ratio as a function of eGFR. Yet, any ratio of two fluctuating variables (here uMMA/Cr) gives a long-tailed distribution. Regression analysis is not suited for such cases because of its sensitivity to outliers. I would suggest to use either logarithmic or square root transformation of Y-axis to suppress the tailing.
(4) Lines 166-174 (Figure 3). Analysis of Fig. 3B and 3C is not justified. The authors cannot correctly assess AUCs for MMA and Hcy, because these variables were used to assign deficiency. Obviously enough, definition of B12 status by plasma MMA (or Hcy) makes this indicator very good to define the status and provides an excellent AUC. I would suggest an alternative approach, where B12-status is assessed by the combined index of B12, MMA, Hcy (DOI 10.1515/cclm-2014-0818) using publicly available XL-sheets. Afterward, the combined index can be used to define B12-deficiency and examine ROC-chart for uMMA/Crn ratio. In such way, panels A, B and C are removed and panel D remains in a modified layout. Anyway, ROC curves for B12, MMA and Hcy are well explored in the literature (e.g. doi: 10.1155/2020/7468506 and references thereof).
Minor comments
Numerous suggestions address the text flow and language issues. They are directly indicated in the submitted file.

Reviewer 2 Report
The manuscript „Diagnostic performances of Urinary Methylmalonic Acid/Creatininuria ratio in vitamin B12 deficiency” pays attention to an important problem: the diagnosis of vitamin B12 deficiency may be difficult in the case of patients with renal impairment since standard biomarkers (increased plasma homocysteine levels and plasma methylmalonic acid levels) may not be specific in this patients. Therefore, the search of the markers independent of renal status seems to be important and useful for clinicians. The aim of the study was to test the diagnostic performances of the urinary methylmalonic acid to urinary creatitine ratio.
The manuscript is generally well written and the Authors characterized study population adequately. The results are clearly presented and thoroughly discussed. The Authors analysed only patients with plasma vitamin B12 201-350 ng/L, which was well justified in the discussion. What is important, they avoid overdiagnosis of vitamin B12 deficiency in case of low level of folate.
In my opinion the article will be interesting for many clinicians.
Round 2
Reviewer 1 Report
General comments
The revised manuscript by Supakul et al “Diagnostic performances of Urinary Methylmalonic Acid/Creatininuria ratio in vitamin B12 deficiency” represents an improved version of the work. Some minor language / misprint issues still remain and require a little more efforts to make the article completely acceptable. I assume that no further proofreading will be required from my side.
Minor comments
(1) Lines 102-104. The definition of deficiency is still nor clear. In the shown context, the authors cannot use the expression “and/or” but either “and” or “or”, depending on their approach to the definition of B12-deficiency. If a patient (e.g. a man below 60) has the following measurements (e.g. B12 = 250 ng/L, Hcy = 25 μmol/L, MMA = 0.25 μmol/L; or alternatively B12 = 250 ng/L, Hcy = 10 μmol/L, MMA = 0.55 μmol/L) and is considered as “deficient”, then the line 104 should run as “… [27]) or ii) plasma …”. If the same person is considered as “not deficient”, then the correct text is “… [27]) and ii) plasma …”. In the case of “and”, the aforementioned B12-deficient individual is supposed to have the dataset including B12 = 250 ng/L, Hcy = 25 μmol/L, MMA = 0.55 μmol/L, where both metabolic markers (MMA and Hcy) are above their respective critical values. In the case of “or”, only one elevated metabolite is sufficient to stipulate the deficiency.
(2) Lines 111-113. Change the text to “The ratio of uMMA/C represents two fluctuating variables, therefore the dependence of y = uMMA/C on x = eGFR was presented after logarithmic transformation (log10(y+1)) of y-axis to suppress deviation of “y” from normality.”
(3) Figure 2B. I assume that the authors intended to write on Y-axis micromol per L (i.e. µM), but not millimol per liter. Please, correct.
(4) Line 159. Add the text. “We should, however, stress, that ROC analysis of the diagnostic performance for plasma MMA and Hcy cannot be considered as accurate, because the same markers were used to stipulate B12-deficiency (thereby considerably adding to their diagnostic value according to AUC).”
(5) A number of suggestions are directly indicated in the submitted file and address corrections of the language and clarification of equivocal expressions.
